# Exploring genotype by environment interaction on cassava yield and yield related traits using classical statistical methods

**Moshood A. Bakare[1]\*, Siraj Ismail Kayondo[2], Cynthia I. Aghogho[2,3], Marnin D. Wolfe[1], Elizabeth Y. Parkes[2], Peter Kulakow[2], Chiedozie Egesi[1,2,4], Ismail Yusuf Rabbi[2], Jean-Luc Jannink[1,5]\***

**1** Plant Breeding and Genetics Section, School of Integrative Plant Science, College of Agriculture and Life Sciences, Cornell University, Ithaca, NY, United States of America, **2** International Institute of Tropical Agriculture (IITA), Ibadan, Nigeria, **3** West Africa Centre for Crop Improvement, University of Ghana, Legon, Ghana, **4** National Root Crops Research Institute Umudike, Umuahia, Nigeria, **5** USDA-ARS Robert W. Holley Center for Agriculture and Health, Ithaca, NY, United States of America

\* jeanluc.work@gmail.com (J-LJ); mab658@cornell.edu (MAB)

**Data Availability Statement:** All data files and analysis scripts are available online on github url address https://github.com/mab658/classical_analysis_GxE.

## Abstract

Variety advancement decisions for root quality and yield-related traits in cassava are complex due to the variable patterns of genotype-by-environment interactions (GEI). Therefore, studies focused on the dissection of the existing patterns of GEI using linear-bilinear models such as Finlay-Wilkinson (FW), additive main effect and multiplicative interaction (AMMI), and genotype and genotype-by-environment (GGE) interaction models are critical in defining the target population of environments (TPEs) for future testing, selection, and advancement. This study assessed 36 elite cassava clones in 11 locations over three cropping seasons in the cassava breeding program of IITA based in Nigeria to quantify the GEI effects for root quality and yield-related traits. Genetic correlation coefficients and heritability estimates among environments found mostly intermediate to high values indicating high correlations with the major TPE. There was a differential clonal ranking among the environments indicating the existence of GEI as also revealed by the likelihood ratio test (LRT), which further confirmed the statistical model with the heterogeneity of error variances across the environments fit better. For all fitted models, we found the main effects of environment, genotype, and interaction significant for all observed traits except for dry matter content whose GEI sensitivity was marginally significant as found using the FW model. We identified TMS14F1297P0019 and TMEB419 as two topmost stable clones with a sensitivity values of 0.63 and 0.66 respectively using the FW model. However, GGE and AMMI stability value in conjunction with genotype selection index revealed that IITA-TMS-IBA000070 and TMS14F1036P0007 were the top-ranking clones combining both stability and yield performance measures. The AMMI-2 model clustered the testing environments into 6 mega-environments based on winning genotypes for fresh root yield. Alternatively, we identified 3 clusters of testing environments based on genotypic BLUPs derived from the random GEI component.

**Funding:** This research was generously funded by the Bill & Melinda Gates Foundation (https://www.gatesfoundation.org) and the United Kingdom's Foreign, Commonwealth & Development Office (FCDO) (https://www.gov.uk/government/organisations/foreign-commonwealth-development-office) through the "Next Generation Cassava Breeding project" (https://www.nextgencassava.org) (Grant INV-007637) managed by Cornell University. Copyright: ©2021 Bakare et al. This is an open access research article distributed and licensed under the Creative Commons Attribution License (CC-BY), that permits unrestricted use, distribution, and reproduction in any medium, provided the original author and source are credited. The funders gave support in the implementation of this study and encourage in publishing this work.

**Competing interests:** The authors have declared that no competing interests exist.

## Introduction

Cassava (*Manihot esculenta* Crantz) is one of the most important food crops worldwide, particularly in sub-Saharan Africa [1,2]. It is known to be a significant source of carbohydrates in the diet of millions of people in developing countries. It is cultivated in diverse edaphic and climatic conditions worldwide [3] due to its efficient carbohydrate production [4] among staple root crops. Cassava is a food security crop grown predominantly by smallholders for subsistence due to its adaptability to survive in drought-prone areas under marginal conditions where other crops may not thrive [1,5]. In comparison to other crops, Sayre et al. [5] reported that cassava is mainly grown under marginal conditions, producing more energy per unit area with limited human input than other crops. Cassava is getting much attention because of its coping mechanisms with diverse environmental conditions [6]. Cassava shows a strong genotype-by-environment interactions (GEI) effects [7], making selection for superior clones a difficult task for cassava breeders. Therefore, selection for a superior clone requires the cassava breeding program to consider the GEI effect. Detailed evaluation of the magnitude and significance of GEI is of utmost importance to ensure greater precision in the release of high yielding and stable clones [7].

Crop phenotypes are well known to be influenced by environmental conditions [8]. This can result in differential genotypic responses across the testing environments resulting in GEI variability. The phenotypic panel for evaluating GEI is often called a multi-environment trial (MET). In METs where genetic lines are often assessed over many years and locations within a target population of environments (TPE), there is usually a critical cross-over interaction (COI), and a GEI term needs to be explored to study the non-additivity of effects.

Among several statistical models devised for exploring the empirical genotypic mean response across environments and for studying and interpreting GEI in agricultural field trials are: Linear models, bilinear models, and linear-bilinear models [9]. Fixed-effect linear-bilinear models such as the Sites Regression (SREG) [10] and the Additive Main Effect and Multiplicative Interaction (AMMI) models [11,12] are used for investigating patterns of genotypic response across environments. In these models, biplots can be used to visualize the patterns of genotypic responses to environments [13,14] that allow the breeders to identify high or low performing clone(s) with broad or specific adaptation for a given trait of interest. A form of the fixed-effect linear model called a factorial regression (FR) model, and a form of the bilinear model, called partial least squares (PLS) regression, allow integrating external environmental and genotypic covariates into the model and can be used to identify weather conditions causing GEI or the genomic segments (e.g., molecular markers) influencing GEI [9].

AMMI is one of the commonly used fixed-effect linear-bilinear models that models the complex structure of GEI. It is a hybrid statistical model combining analysis of variance (ANOVA) to model main effects of genotype and environment and principal component analysis (PCA) to decompose complex GEI structure into Interactive Principal Component Axes (IPCAs) through singular value decomposition. In this model, the percentage of GEI variation explained by IPCAs decreases, with the first IPCA accounting for the highest percentage of GEI variation. The AMMI biplot of first IPCA scores against the mean of genotypic performance visualizes both genotypes and environments through which genotypes with broad or specific adaptation can be identified. Genotypes with IPCA score in the vicinity of zero are considered stable across environments.

However, genotypes with scores that deviate from zero for a given IPCA are unstable relative to the determinants of that IPCA but may exhibit specific adaptation if they are identified as close to a particular environment in the AMMI biplot. AMMI is often preferable to a linear regression approach in the sense of being parsimonious as it requires fewer degrees of freedom

to explain GEI. The AMMI model can further delineate the testing environments with the best genotypes into mega environment using principal component axes scores and AMMI stability values (ASV) [15]. The AMMI stability value (ASV) for a genotype is defined to be the distance from the coordinate point for that genotype to the origin in a two-dimensional space of the first two Interactive principal component analysis scores (IPCA1 and IPCA2, [16]. Because IPCA scores account for different amounts of variation in the GEI sum of squares a weighted value must be assigned to assess stability using the AMMI model. Genotypic stability alone does not provide a sufficient yardstick for selection as stable genotypes might not necessarily yield the highest yield performance. Mahmodi et al. [17] and Tumuhimbise et al. [2] used a genotype selection index (GSI) which is sum of genotypic yield rank across environments and ASV rank to identify high yielding and stable genotypes. This index implicitly values yield and stability equally. A low GSI value signifies a desirable genotype with high average yield performance and high stability [17].

The Site Regression (SREG) model, also called Genotype Main Effect plus Genotype-Environment Interaction (GGE), is a modification of the AMMI model where the bilinear term combines the genotype main effect (G) and the GEI effect in a multiplicative term. It allows breeders to explore total genetic rather than exclusively GEI variation. The GGE model enables the finding of GEI in terms of crossover resulting from changes in genotypic ranking across the environments [18]. Unlike AMMI biplots that approximate only GEI, the genotypic scores in a GGE model describe the G and GEI jointly to approximate overall performance of (G + GE) in environments.

This study's principal objectives were: (i) To identify stable and high-yielding cassava clones adapted to broad and/or specific environments; (ii) To determine the relative importance of sources of variation influencing key agronomic traits; and (iii) To identify mega environments for Nigerian cassava; iv) To provide a clear road map for other researchers pursuing these types of objectives.

## Materials and methods

### Clonal material and experimental field design

Thirty-six advanced cassava clonal lines were evaluated (31 experimental lines and five standard checks, Table 1) as in the uniform yield trial (UYT), an advanced evaluation stage of the International Institute of Tropical Agriculture (IITA) cassava breeding program. The clones were derived from diverse parentage of elite x elite crosses combination part of genomic recurrent breeding program. The clones were selected for further field assessment after four stages of screening for diseases, vigor, agronomic and quality traits of interest in early stages of breeding program. They were resistant to diseases and had potential for high yielding and dry matter content. These clones were evaluated across 20 trials grown in 11 locations across different agro-ecological zones in Nigeria (Fig 1) over three cropping seasons (2017–2018, 2018–2019, and 2019–2020).

Weather data were collected from the database of the National Aeronautics and Space Administration Prediction of Worldwide Energy Resource project (https://power.larc.nasa.gov/data-access-viewer/). Average minimum and maximum temperature of the testing environments over crop growth cycle ranged from 19˚C for Zaria20 to 24˚C for Onne19. As for average maximum temperature, it varied from 29˚C for Onne19 to 36˚C for Kano19. Meanwhile, the total precipitation varied between 415 mm for Kano19 and 5395 mm for Onne20 with average relative humidity of approximately 41% and 89% respectively (Fig 2).

Each trial was established as a Randomized Complete Block Design (RCBD) in three replicates. The experimental plot consisted of six rows of length 5.6 m with an inter-row spacing of

**Table 1. Thirty-six elite cassava clonal lines evaluated across 11 locations in Nigeria over three cropping seasons.**

| Clone | Pedigree | Cycle | Clone Year |
|---|---|---|---|
| IITA-TMS-IBA000070 (Check) | TMEB459 X? | C0 | 2000 |
| IITA-TMS-IBA30572 (Check) | 58308 X BRANCA DE SANTA CATARINA | C0 | 1973 |
| IITA-TMS-IBA980581 (Check) | NA | C0 | 1998 |
| IITA-TMS-IBA982101 (Check) | IITA-TMS-IBA951181 X IITA-TMS-IBA71173 | C0 | 1998 |
| TMEB419 (Check) | NA | C0 | NA |
| TMS13F1021P0008 | IITA-TMS-IBA010903 X IITA-TMS-IBA030075 | C1 | 2013 |
| TMS13F1114P0001 | IITA-TMS-IBA070126 X IITA-TMS-IBA000355 | C1 | 2013 |
| TMS13F1182P0002 | IITA-TMS-IBA011412 X TMEB419 | C1 | 2013 |
| TMS13F1461P0002 | IITA-TMS-MM990268 X IITA-TMS-IBA000355 | C1 | 2013 |
| TMS13F2061P0005 | (IITA-TMS-IBA070004 X IITA-TMS-IBA070520 X SM3361-30)-11 | C1 | 2013 |
| TMS13F2207P0001 | IITA-TMS-KAN930061 X IITA-TMS-IBA960249 | C1 | 2013 |
| TMS14F1001P0004 | TMS13F1303P0001 X TMS13F1020P0002 | C2 | 2014 |
| TMS14F1016P0006 | TMS13F1307P0011 X TMS13F1108P0007 | C2 | 2014 |
| TMS14F1022P0006 | TMS13F1307P0020 X TMS13F1106P0006 | C2 | 2014 |
| TMS14F1035P0004 | TMS13F1095P0009 X TMS13F1307P0008 | C2 | 2014 |
| TMS14F1035P0007 | TMS13F1095P0009 X TMS13F1307P0008 | C2 | 2014 |
| TMS14F1036P0007 | TMS13F1109P0009 X TMS13F1307P0020 | C2 | 2014 |
| TMS14F1049P0001 | TMS13F1391P0039 X TMS13F1306P0003 | C2 | 2014 |
| TMS14F1120P0003 | TMS13F1309P0001 X TMS13F'1333P0003 | C2 | 2014 |
| TMS14F1131P0001 | TMS13F1087P0002 X TMS13F1176P0003 | C2 | 2014 |
| TMS14F1194P0002 | TMS13F1101P0007 X TMS13F1307P0020 | C2 | 2014 |
| TMS14F1195P0005 | TMS13F1106P0006 X TMS13F1307P0020 | C2 | 2014 |
| TMS14F1208P0007 | TMS13F1106P0006 X TMS13F1020P0002 | C2 | 2014 |
| TMS14F1223P0007 | TMS13F1106P0006 X TMS13F1108P0007 | C2 | 2014 |
| TMS14F1224P0004 | TMS13F1106P0006 X TMS13F1212P0055 | C2 | 2014 |
| TMS14F1262P0002 | TMS13F1063P0009 X TMS13F1307P0008 | C2 | 2014 |
| TMS14F1285P0017 | IITA-TMS-IBA961632 X IITA-TMS-IBA000070 | C2 | 2014 |
| TMS14F1291P0011 | IITA-TMS-IBA030055A X IITA-TMS-IBA961632 | C2 | 2014 |
| TMS14F1297P0019 | IITA-TMS-IBA020431 X IITA-TMS-MM970806 | C2 | 2014 |
| TMS14F1300P0008 | IITA-TMS-ZAR930151 X IITA-TMS-MM970043 | C2 | 2014 |
| TMS14F1303P0012 | I IITA-TMS-ZAR930151 X ITA-TMS-IBA930134 | C2 | 2014 |
| TMS14F1306P0015 | IITA-TMS-IBA030060 X IITA-TMS-MM970043 | C2 | 2014 |
| TMS14F1306P0020 | IITA-TMS-IBA030060 X IITA-TMS-MM970043 | C2 | 2014 |
| TMS14F1310P0004 | IITA-TMS-IBA030060 X IITA-TMS-IBA930265 | C2 | 2014 |
| TMS14F1311P0020. | IITA-TMS-IBA030060 X IITA-TMS-ZAR930151 | C2 | 2014 |
| TMS14F1312P0003 | IITA-TMS-IBA930134 X IITA-TMS-ZAR930151 | C2 | 2014 |

1 m and intra-row spacing of 0.8 m. The locations used varied from one cropping season to another as did the number of trials, resulting in an unbalanced data structure.

## Statistical analysis

**Data quality control and single trial analysis.** Before formal genotype-by-environment analysis, the empirical distribution of the observed agronomic traits was visualized by unique environment (i.e., location by year combination) in boxplots using the ggplot2 package [19] in R [20]. A linear mixed model was fitted to the individual trials to estimate clonal variance components and broad-sense heritability. The Proc Mixed procedure of Statistical Analysis

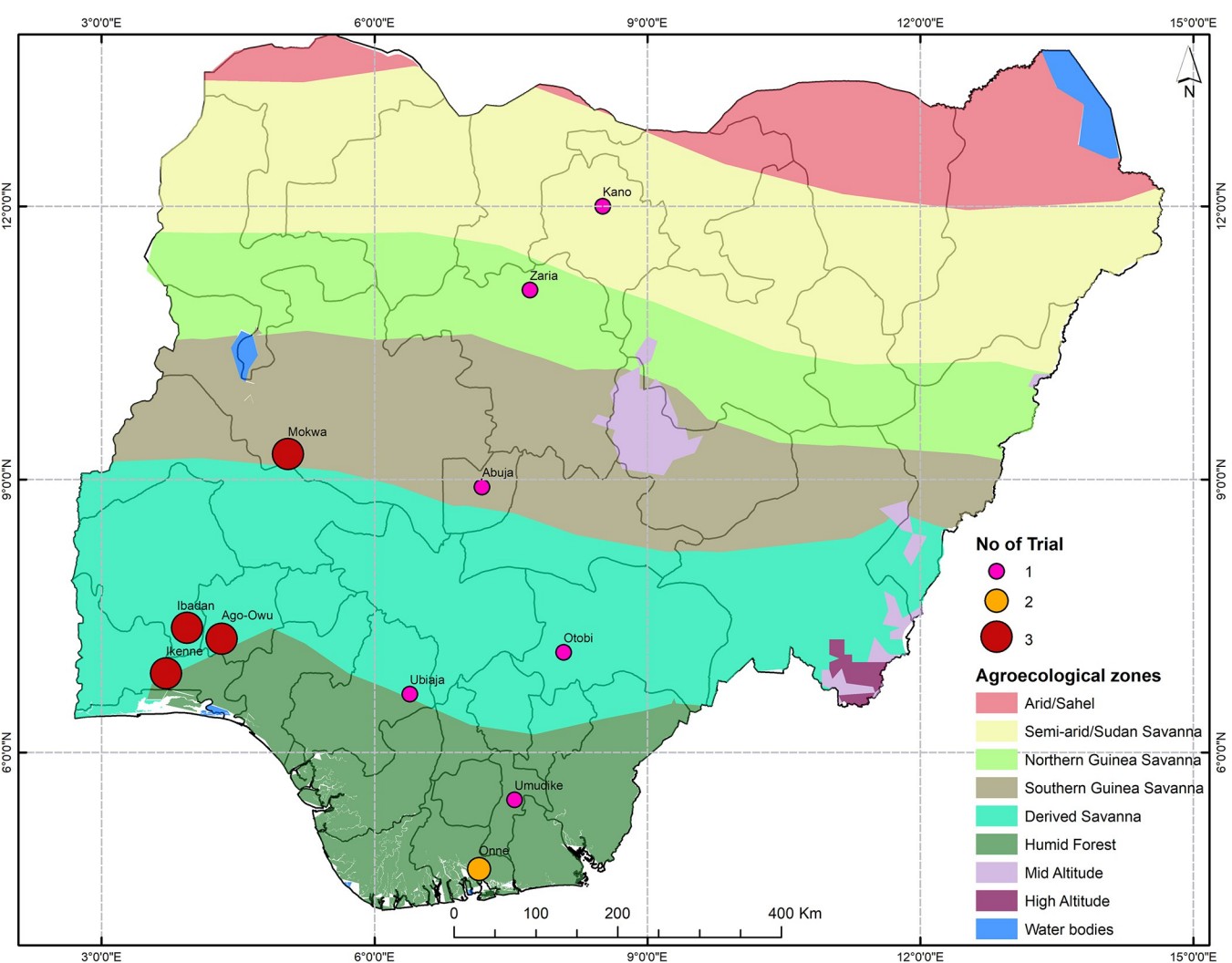

**Fig 1. A map of Nigeria showing the field trial locations across agro-ecological zones.**

Software (SAS) software version 9.4 [21] was used to fit the following model:

$$y = \mu + X_1 r + p\beta + Z_1 g + \epsilon \tag{1}$$

where $y$ is the vector (n × 1) of observed phenotypic values, in which $n$ is the number of observations; μ is the intercept (overall mean); $r$ is the vector (i × 1) of fixed effects of ith replicates with its associated incidence matrix $X_1$; $p$ denotes the proportion of plant stands harvested, a covariate for all traits except dry matter content; $\beta$ is a regression coefficient relating $p$ and $y$; $g$ is the vector (j × 1) of random effects of jth genotype with its associated design matrix $Z_1$, and $\in$ is a residual term assumed to follow a Gaussian distribution.

The quality of each trial was assessed by calculating the coefficient of variation (CV), broad-sense heritability (H2), and experimental accuracy (Ac) proposed by [22] using the following expressions: $\mathrm{CV\%} = (\hat{\sigma}_e / \bar{y}) \times 100$, $\mathrm{H}^2 = \hat{\sigma}^2_g / (\hat{\sigma}^2_g + \hat{\sigma}^2_e)$, and $\mathrm{Ac} = \sqrt{(1 - \mathrm{PEV}/\hat{\sigma}^2_g)}$ in which $\hat{\sigma}_e$ is the estimated residual standard deviation, $\bar{y}$ is the estimated overall mean for a trait; $\hat{\sigma}^2_g$ is the estimated genetic variance $\hat{\sigma}^2_e$ is the estimated error variance, and PEV is the average of prediction error variance.

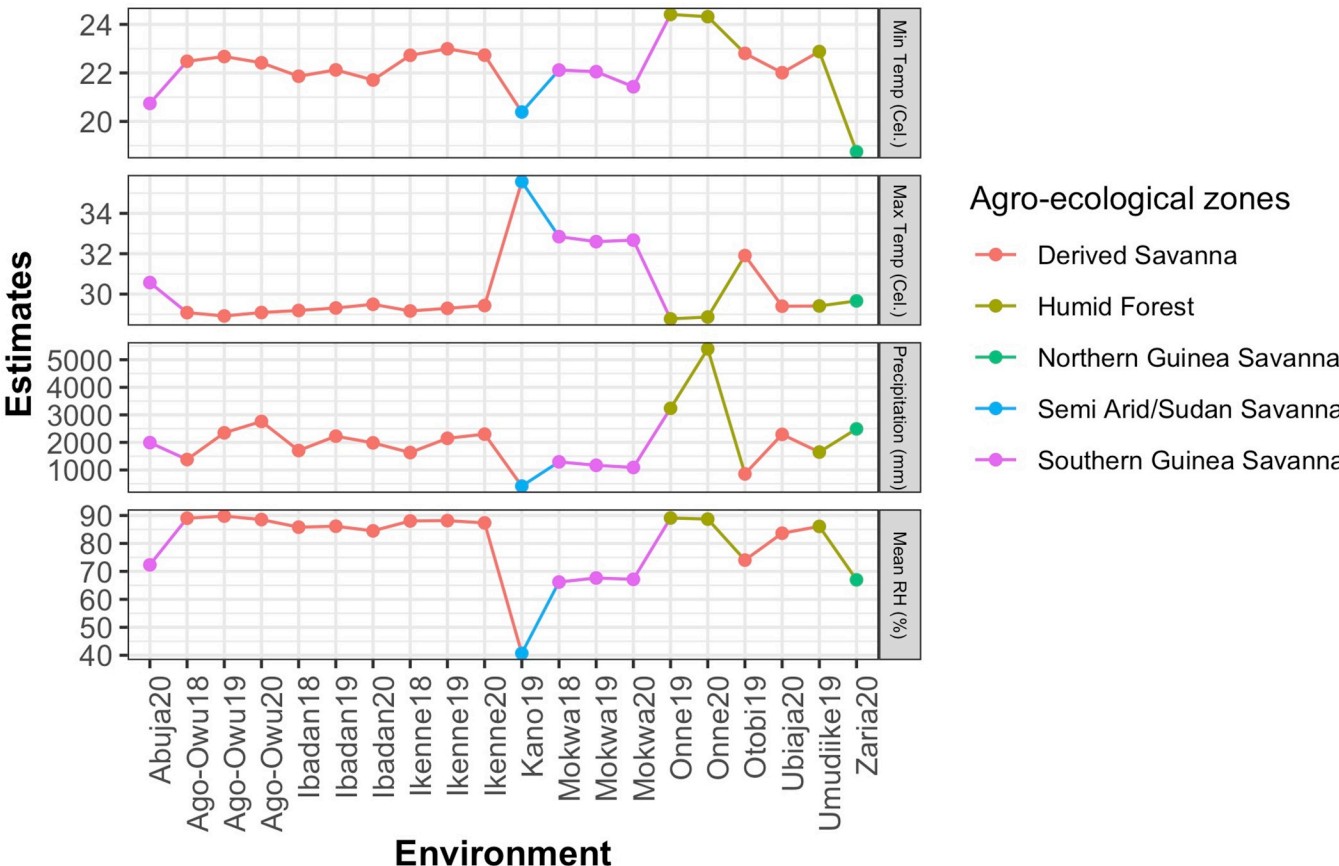

**Fig 2. Trends of average minimum and maximum temperature (˚C), total precipitation (mm), and mean relative humidity across the testing environments (%).**

This analysis identified three trials with low heritability ($H2 < 0.1$), low accuracy ($Ac < 0.4$) and high CV ($CV > 40.5$). These trials were removed from further analysis.

**Joint G×E analysis of multiple trials.** We then carried out a combined linear mixed model analysis on data consisting of $g$ genotypes evaluated across $e$ environment in $r$ replicates within each environment. The model fitted for each agronomic trait was:

$$y_{ijk} = \mu + g_i + e_j + b_{k(j)} + ge_{ij} + p\beta + \epsilon_{ijk} \tag{2}$$

where $y_{ijk}$ is a phenotypic vector of the observed agronomic trait of ith genotype in kth replicate within jth environment; $\mu$ is a fixed intercept, $g_i$ is the random effect of ith genotype, $e_j$ is the random effect of jth environment, $b_{k(j)}$ is the random block effect within jth environment, $ge_{ij}$ is the random interaction effect of ith genotype and jth environment, $p$, $\beta$, and $\in_{ijk}$ were as defined in the previous equation.

The random effects in the model are postulated to follow a multivariate normal distribution with means and variances defined as:

$$g \sim N\left(0, I\sigma^2_g\right), \ e \sim N\left(0, I\sigma^2_e\right), \ b \sim N\left(0, I\sigma^2_b\right), \ ge \sim N\left(0, I\sigma^2_{ge}\right), \ \text{and} \in \sim N\left(0, I\sigma^2_\in\right)$$

where 0 is the expected value (mean) of zero; $\sigma2_g$ is the genetic variance; $\sigma2_e$ is the environmental variance; $\sigma2_b$ is the block variance nested with jth environment; $\sigma2_{ge}$ is the variance of genotype-by-environment interaction $\sigma2_\in$ is the residual variance; and $I$ is the identity matrix,

with order equal to the number of observations. We calculated the percentage of total phenotypic variance explained by each random effect to determine how significant it influenced each trait's variability. Broad-sense heritability on plot mean basis across all environments was derived from variance components estimate as

$$H^2 = \frac{\sigma_g^2}{\sigma_g^2 + (\sigma_{ge}^2/e) + (\sigma_\epsilon^2/er)} \qquad (3)$$

where $e$ is the number of environments, $r$ is the number of replicates of genotypes per environment, and other terms were described above. Out of 17 trials, 12 had three replicates and the others had two replicates. Therefore, the harmonic mean was calculated to be approximately 2.6, which was used as the number of replicates across all the trials to compute the heritability.

We further ascertained the presence or absence of GEI by fitting both the reduced model without the GEI term and a full model that included the GEI term. The likelihood ratio test (LRT) was carried out on each of the agronomic traits to determine if there was a significant improvement in fitting a full model. In the same manner, we tested for the homogeneity versus heterogeneity of error variance across trials. Finally, we further partitioned the GEI variance into a repeatable component as genotype by location (GL), and non-repeatable components as genotype by year (GY) and genotype by location by year (GLY). In the presence of significant GEI, we assessed its pattern by fitting Finlay Wilkinson (FW), Additive main effect and multiplicative interaction model (AMMI), and genotype and genotype by environment (GGE) models on the two-way genotype environment adjusted means using the statgenGxE package [23] in R [20], as described below.

**Finlay-Wilkinson regression.** The Finlay-Wilkinson regression approach [24] was used to model GEI by regressing mean phenotypic performance of individual genotypes on an environmental index and determine the heterogeneity of associated slopes. The index value of an environment was the mean of all clones for the trait in that environment. This method requires two steps: (i) Compute the index values, and (ii) Estimate intercept and slope for each genotype by regressing genotypic performance on the environmental index. Prior to fitting the Finlay Wilkinson model, trait values were scaled to mean of zero and standard deviation of one following the equation below to allow comparison of means square error (MSE) values across traits, which are measured in different scale of units as:

$$y_{ij_{standardized}} = [y_{ij} - mean(Y)]/sd(Y) \qquad (4)$$

where $y_{ij}$ is the adjusted phenotypic mean value of ith genotype in jth environment and $Y$ is the overall mean of adjusted phenotypic response of all clones in all environments. This standardization of each trait necessitated that the MSE values reflected variability and not the absolute scale of a given unit [25]. Then for each trait, we fitted the Finlay-Wilkinson model as

$$y_{ij} = \mu + g_i + \beta_i e_j + \epsilon_{ij} \qquad (5)$$

in which $y_{ij}$ is as described above but scaled, $\mu$ is overall mean, $g_i$ is the genotypic intercept, $\beta_i$ is a slope representing the sensitivity of ith genotype. The average value of $\beta_i$ is 1; $\beta_i > 1$ shows $i$ has higher than average sensitivity, and $\beta_i < 1$ shows $i$ has lower than average sensitivity [26], $e_j$ is the environment sample mean, and $\epsilon_{ij}$ is a random error term associated with ith genotype evaluated in jth environment.

The analysis provides three key parameters for each genotype: the intercept $g_i$, which expresses the general performance of a genotype, the slope $\beta_i$, which measures the sensitivity of a genotype, and the residual variance $var(\epsilon_{ij} \mid i)$, which is a deviation from the regression line

denoting the stability. To quantify and compare trait sensitivity to GEI, the variance of slope and that of MSE resulting from Finlay-Wilkinson model were used [25].

**AMMI analysis.** The observed traits' G×E interaction was analyzed using the Additive Main effect and Multiplicative interaction (AMMI) model. AMMI is a fixed effect linear-bilinear model which analyses the main effect of genotype and environment using ANOVA and the multiplicative effect using principal component analysis (PCA) in a single model [27]. Each of the agronomic traits was subjected to AMMI analysis by fitting the model

$$y_{ij} = \mu + g_i + e_j + \sum_{k=1}^{K} (\lambda_k \alpha_{ik} \gamma_{jk}) + \epsilon_{ij} \tag{6}$$

where $y_{ij}$ is the mean performance of ith genotype in jth environment; $\mu$ is the intercept; $g_i$ is fixed effect of ith genotype; $e_j$ is the fixed effect of jth environment. The GEI component is decomposed into K multiplicative terms (k = 1, 2, . . ., K), each multiplicative term is a product of the kth eigenvalue ($\lambda_k$); genotypic score ($\alpha_{ik}$); and environmental loadings ($\gamma_{jk}$); and $\epsilon_{ij}$ is the residual GEI not captured by the model and some error deviation.

We computed AMMI Stability Value (ASV) for each genotype relative to the influence of IPCA1 and IPCA2 scores based on their interaction sum of squares according to Purchase [16] using the formula:

$$ASV = \sqrt{\left[ \left( \frac{SS_{IPCA1}}{SS_{IPCA2}} \right) \times IPCA1 \right]^2 + IPCA2^2} \tag{7}$$

where ($SS_{IPCA1}/SS_{IPCA2}$) was the weight assigned to the *IPCA1* value by dividing the *IPCA1* SS by the *IPCA2* SS; and the *IPCA1* and *IPCA2* scores were the genotypic score derived from the AMMI model. A large positive ASV value indicates a genotype that is adapted to particular environments. A small (close to zero) ASV value indicates a stable genotype across environments [16].

We also calculated a genotype selection index (GSI) for each genotype as the sum of genotypic rank based on mean yield across environments (RY) and rank of AMMI stability value (RASV):

$$GSI_i = RASV_i + RY_i \tag{8}$$

The genotype with the lowest GSI value is considered the most valuable [28].

**GGE analysis.** Genotype main effect and Genotype by Environment (GGE) analysis is a modification of AMMI analysis. Unlike AMMI, only the environment is fitted as a main effect in the GGE model. This brings about fitting principal component analysis jointly on genotype main effect and genotype by environment interaction as a sum of multiplicative terms. The GGE analysis does the job of fitting the principal component model with two components to the two-way genotype by environment table of mean centered per environment with genotypes as object and environments as variable [18]. Like AMMI, the principal component scores can be exploited in constructing biplots. The GGE model or AMMI analysis could be used to define mega-environments [29]. The observed traits were subjected to GGE analysis by fitting GGE model as

$$y_{ij} = \mu + e_j + \sum_{k=1}^{2} (\lambda_k \alpha_{ik} \gamma_{jk}) + \epsilon_{ij} \tag{9}$$

where each term is as described for the AMMI model.

**Mega-environment delineation.**   In the context of GEI, a mega-environment is defined to be a group of environments sharing a common best performing genotype. In principle, it also follows that different genotypes are adapted to different mega-environments and GEI variation between the mega-environments is higher than variation within the mega-environments [30]. We determined mega-environments based on the AMMI-2 model of order 2. The environments were clustered using the *gxeMegaEnv()* function of the statgenGxE package [23] based on the fitted values from AMMI-2 model. Environments that share the common best genotype belong to the same mega-environment.

The version of GGE biplot graphic called "which won where" plot is also a tool for the delineation of a mega-environment. In the case of delineating mega-environment through GGE biplot analysis, the resulting mean value in the graphics is related to mega-environment mean and not grand mean, and it supports in identifying genotypes with broad or narrow adaptation to some environments or groups of environments [31]. The "which won where" biplot includes an irregular polygon whose vertices mark the genotypes that are furthest from the origin in all directions such that the polygon encompasses all genotypes in the biplot. Lines are also drawn originating from the biplot's origin and intercepting the polygon's sides perpendicularly [32]. The lines emanating from the origin split the biplot into sections and the genotype at the vertex of every section had the optimal yield performance in environments contained in that section. Each section in effect defines a mega-environment.

**Cultivar superiority index.**   Further assessment of the stability of each clone was determined after testing the significance of GEI. We quantified yield stability across the testing environments using a univariate stability estimate called cultivar-superiority measure [33]. It is a measure of stability by superiority index, and it is defined as a function of the sum of the squared differences between a cultivar's mean performance and the best cultivar's mean, where the sum is across trials. Lin and Binns [33] proposed the calculation of superiority index using expression:

$$P_i = \sum_{j=1}^{n} (X_{ij} - M_j)^2 / 2n \tag{10}$$

where $P_i$ is the superiority index of ith cultivar; $X_{ij}$ is the yield of ith cultivar in the jth environment; $M_j$ is the highest yield response got among the cultivars in the jth environment; and $n$ is the number of environments. This expression was further decomposed as

$$P_i = [n(\bar{X}_{i.} - \bar{M})^2 + \sum_{j=1}^{n} (X_{ij} - M_j + \bar{M})^2] / 2n \tag{11}$$

where $\bar{X}_{i.} = \Sigma n_{j=1} X_{ij}/n$, and $\bar{M} = \Sigma n_{j=1} M_j/n$, $\bar{X}_{i.}$ = mean yield of ith cultivar in $n$ environments and $\bar{M}$ = mean of maximum response in the $n$ environments. According to Lin and Binns [33], the first term of $P_i$ quantifies genetic variation and the second term quantifies GEI. Cultivars with the lowest values of the index $P_i$ are more stable and close to the best cultivar in each environment.

**Representative of target population of environments.**   We considered all environments in the study to be the target population of environments (TPE). We identified testing environments that best represented the TPE by following these steps: i) calculate environment-specific genotypic BLUPs by fitting genotype effect as random, ii) calculate genotypic BLUPs across all environments which represent a TPE, iii) calculate the Pearson correlation between environment-specific genotypic BLUPs and genotypic BLUPs across all environments as a measure of breeding value accuracy, and iv) estimate environment-specific heritability based on the Cullis approach which involves the variance of a difference between genotypes. Cullis et al. [34]

proposed to compute heritability as

$$H_{Cullis}^2 = 1 - \frac{\bar{V}_{\Delta..}^{BLUP}}{2\sigma_g^2} \tag{12}$$

where $\bar{v}_{\Delta..}{}^{BLUP}$ is the mean variance of a difference of two genotypic BLUPs and $\sigma 2_g$ is the genetic variance, and v) rank heritability and Pearson correlation value and take the sum of their rank. We use both high genetic correlation and heritability estimate as indicators for identifying a good representative for the TPE.

To determine the number of testing environments representing the entire TPE, we randomly sampled subsets of 1 to 16 environments from the phenotypic data repeatedly for 50 times. For each sampling, a model was fitted to obtain genotypic best linear unbiased prediction (BLUP). Then, for each sampling environment, Pearson correlation was obtained between the BLUP and the BLUP derived from all the environments. We further calculated the average correlation coefficient as a breeding value accuracy relative to overall environments. The point at which the line plot showing the trends of breeding values accuracy relative to all TPE, and sampled environments reaches a plateau was used to determine optimal number of environments to represent TPE.

To provide further insights into the relatedness or grouping of the current testing environments based on key traits, we extracted environment-specific genotypic BLUPs from the random G×E effect component of the joint analysis. Then, we further carried out a Pearson correlation analysis among the environments. Thereafter, the clustering of the environments was carried out based on a distance matrix derived from correlation matrix using ward.D2 linkage method [35]. Intuitively, we examined the resulting dendrogram from the clustering to identify environments that joined together with the smallest distance as a cluster group.

## Results

### Phenotypic data description and single trial analysis

The distribution of the phenotypic values of observed traits of the 36 clones revealed that all observed traits approximated a normal distribution across the testing environments satisfying the assumption of normality in classical statistical methods (Fig 3). We observed a range in variation of fresh root yield (t/ha) from low performance environments (Onne19, Onne20, Ubiaja20), to high performance environments (Ibadan19, Otobi19, Ago-owu20, Ikenne20). The boxplots further revealed the heterogeneity of variability for the observed traits across the environments indicating the presence of GEI.

The plots resulting from data quality control of single-trial analysis (S1 and S2 Figs) showed that the three trials: 18UYT36setAKN, 19UYT36setAZA, and 19UYT36setAMK should be removed based on thresholds set for CV, H2, and Ac. The mean fresh root yield across the remaining 17 trials ranged from 6.52 t/ha (19UYT36setAON) in Onne20 to 47.49 t/ha (18UYT36setAOT) in Otobi19 with an overall mean of 27.46 t/ha (S1 Table). The summary statistics for other traits like dry matter content, dry yield, top yield, and harvest index from each testing environment are also reported (S1 Table). In addition, visualization of the distribution of derived parameters such as broad-sense heritability, coefficient of variation, experimental accuracy, and residual variance for all traits across all trials were showed (S3 Fig). We observed smallest variability in dry matter content and then harvest index across all trials relative to other traits.

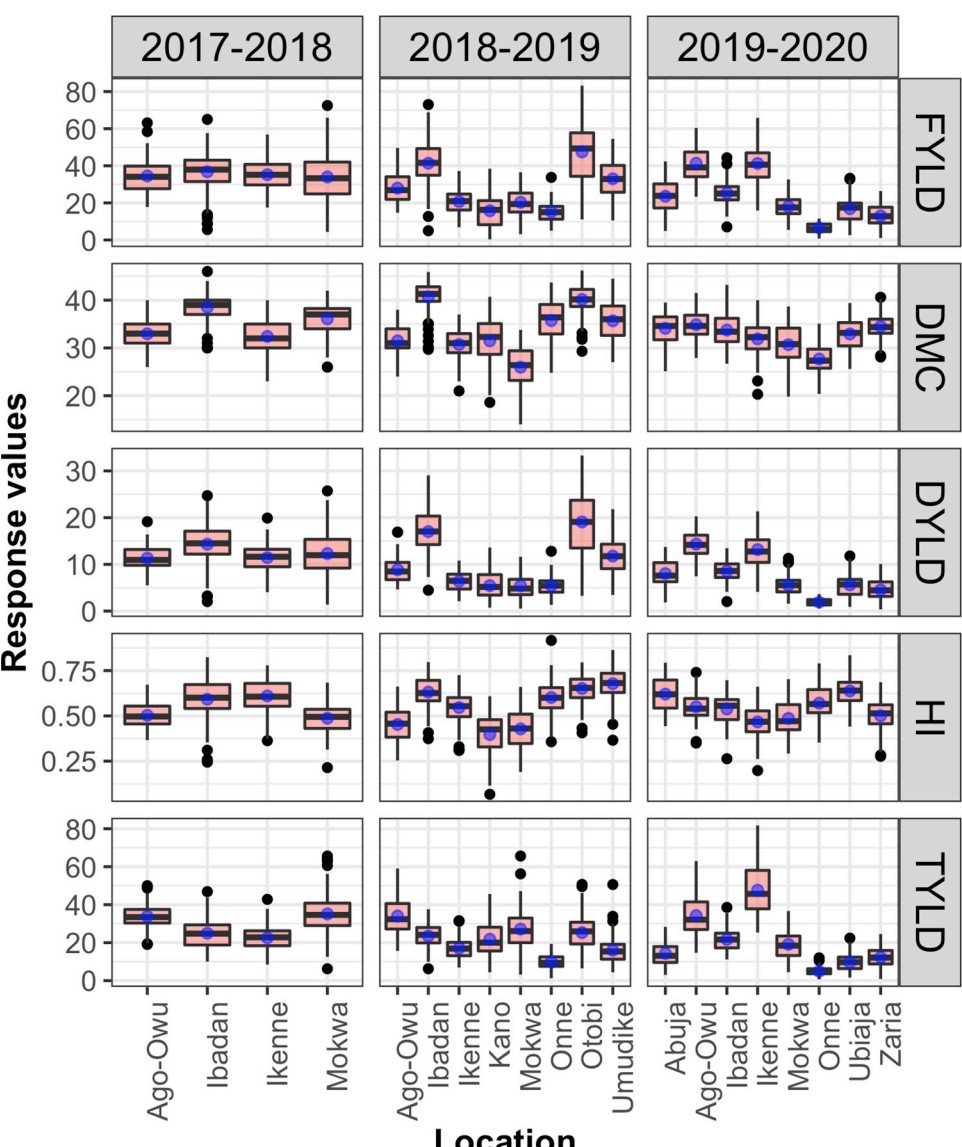

**Fig 3. The distribution of fresh root yield (FYLD t/ha), dry matter content (DMC %), dry yield (DYLD t/ha), harvest index (HI), and top yield (TYLD t/ha) of 36 clones evaluated across 20 environments.**

## Combined analysis of multiple trials

The likelihood ratio test (LRT) statistics identified presence of significant GEI and error variance heterogeneity across testing environments for all observed traits (S2 and S3 Tables). The percentage of phenotypic variance attributed to each model term for each trait was reported (Fig 4 and S4 Table). The environment had a significant effect on all traits (P < 0.01) and captured the largest percentage of total phenotypic variance ranging from 48.6% in harvest index to 63.9% in top yield. The genotypic effect was highly significant (P < 0.001) for each trait and explained a percentage of phenotypic variance between 2.6% (harvest index) and 12.6% (dry matter content). The GEI term was also highly significant (P < 0.001) and accounted for 5.3% (top yield) to 12.5% (harvest index) of phenotypic variance. We observed relatively high GEI variance compared to genetic variance for fresh root yield. In contrast, genetic variance for top

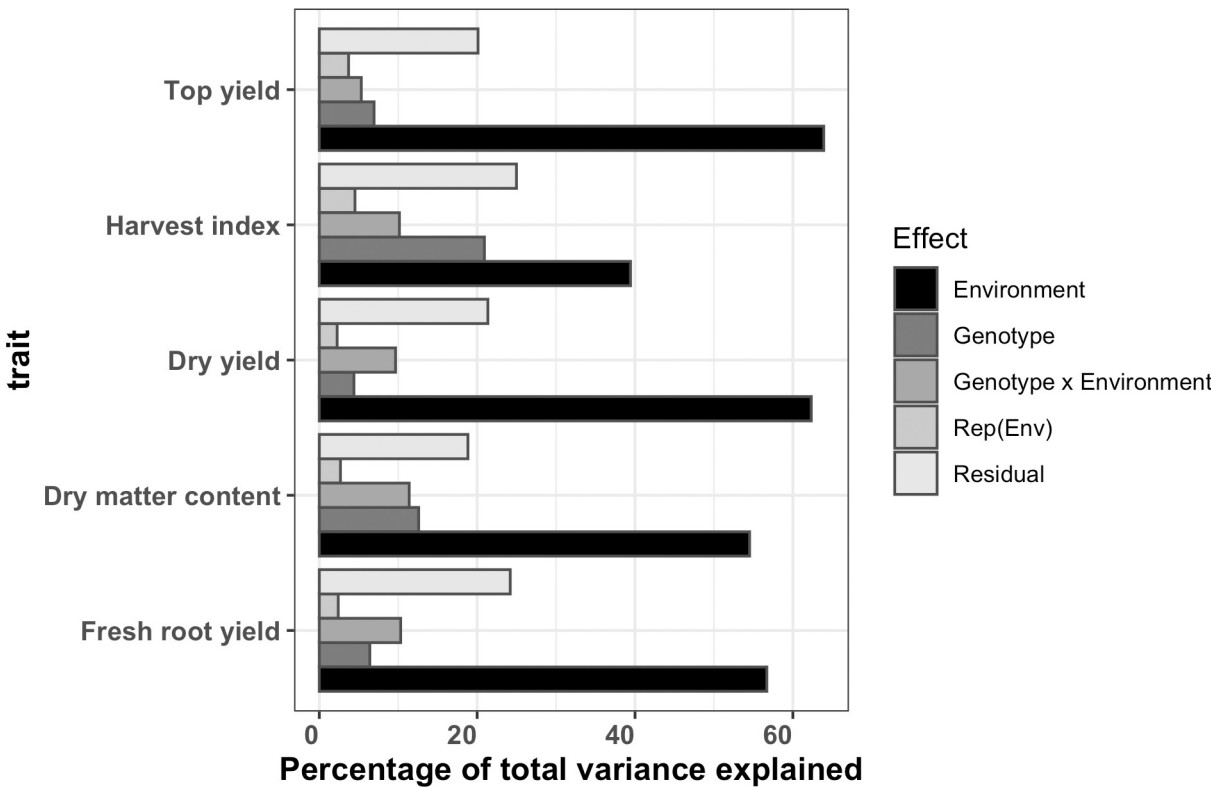

**Fig 4. The percentage of total phenotypic variance attributed to each effect for each trait across 17 trials.**

yield was higher than GEI variance indicating environmental conditions played a lesser role in influencing top yield (Fig 4 and S4 Table). The replication nested within environment captured between 2.3–5.6% of the phenotypic variance, which was the smallest relative to other source of variation. It was highly significant (P < 0.01) for all traits. The residual term was the second largest source of variation after the environment effect. It accounted between 18.8–30.8% of the phenotypic variance (Fig 4 and S4 Table). The broad-sense heritability estimates varied from 0.64 for harvest index to 0.92 for dry matter content (S4 Table).

Further decomposition of GEI term into repeatable component (GL) and non-repeatable component (GY and GLY) revealed that GY component was not significant for all traits except dry matter content (P < 0.001). It accounted for the smallest portion of phenotypic variance ranging from approximately 0.2 to 5.6% (S5 Table). Repeatable GL component explained between 3.7–48.6% percentage of phenotypic variance. In comparison to GL and GY components, the GLY term was highly significant (P < 0.001) and accounted for largest portion of phenotypic variance for all traits except for harvest index where GL explained largest portion of phenotypic variance. The significance of GLY is an indication that for all traits, genotypic response to conditions particular to a specific location depends on year of evaluation and vice versa.

## Finlay-Wilkinson regression

The genotypic and environmental main effects of the Finlay-Wilkinson (FW) model were highly significant (P < = 0.001) for all observed traits (S6 Table). Significant differences in regression slope (sensitivity) among genotypes on the environmental mean was found for all

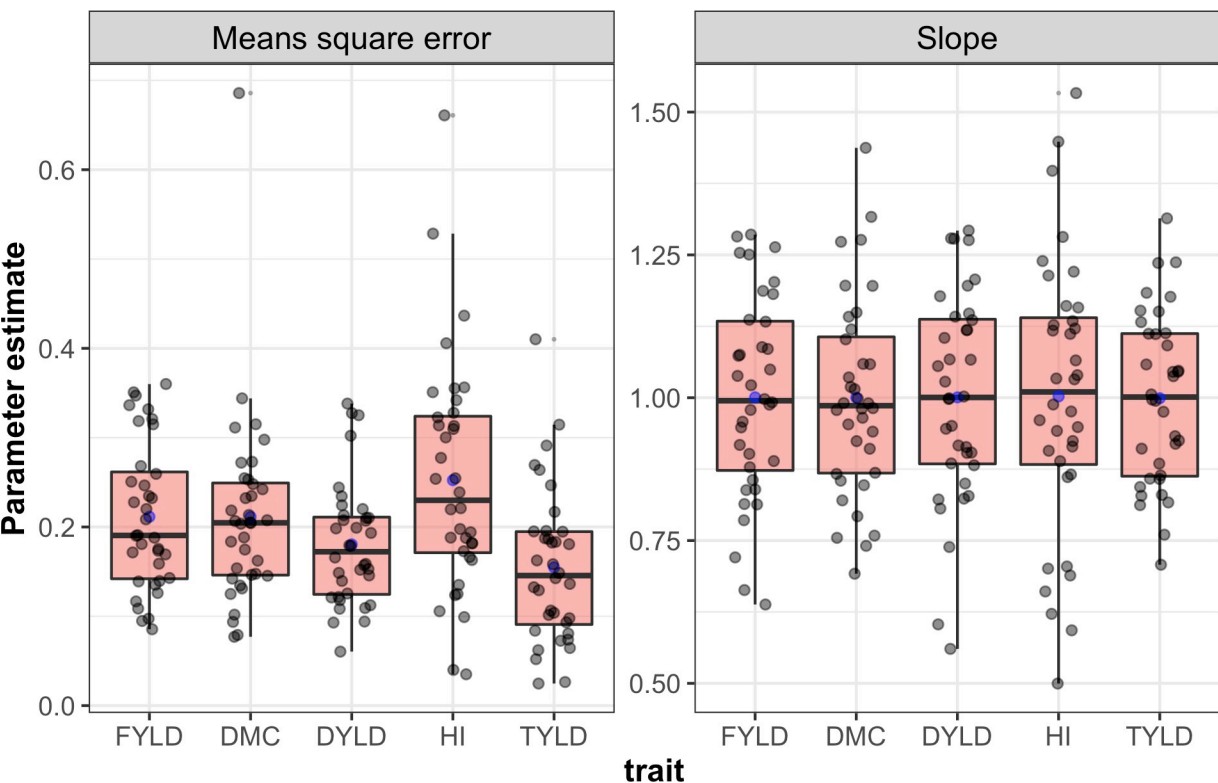

**Fig 5. The distribution of MSE and slope resulting from Finlay Wilkinson model for the evaluation of 36 elites' cassava clones across 17 environments for five traits.**

traits except dry matter content (S6 Table). In other words, there was variation in genotypic response for all traits except dry matter with respect to changes in environment mean.

The genotypic sensitivity values were ranked from the most stable (low sensitivity value) to the least stable (high sensitivity value) for each trait (S7 Table). The FW model identified TMS14F1297P0019, TMEB419, TMS14F1120P0003, TMS13F1461P0002, and TMS14F1312P0003 as the top 5 most stable genotypes for fresh root yield with sensitivities values of 0.638, 0.663, 0.721, 0.786, and 0.813 respectively.

Trait sensitivity to GEI was quantified by the variance of the slopes and the variance of MSEs. The top yield had the lowest median MSE of all traits (median = 0.146) and the variance of MSE (variance = 0.007) (Fig 5 and S8 Table). Meanwhile, the slope variance ranged from 0.023 (top yield) to 0.058 (harvest index) with the corresponding slope median values of 1.001 and 1.010 respectively, (Fig 5 and S9 Table).

## AMMI analysis

The AMMI analysis revealed significant variation in the main effects of genotype (G), environment (E) and their interactions (GEI) (P < 0.001) for all observed traits (S10 Table). The partition of total sum of squares (TSS) showed that the environment main effect accounted for highest amount of variation varying from 48.2% (harvest index) to 76.1% (top yield).

The decomposition of variation in GEI for fresh root yield showed that the first and the second interactive principal components (IPCs) captured 21.6% and 15.7% and accounted for 4.5% and 3.3% of the TSS. The first two IPCs accounted for 21.0% and 15.9% of GEI SS and 4.3% and 3.3% of TSS for dry matter content. Finally, the partition of variation in GEI for top

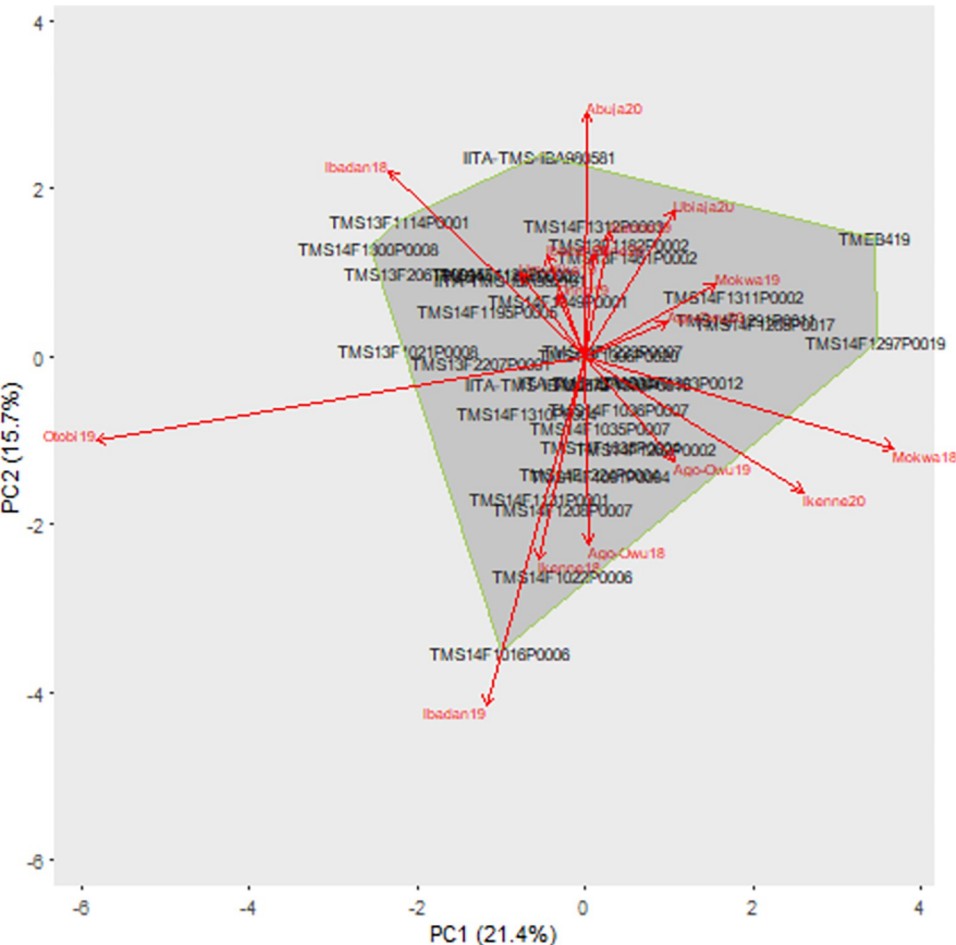

**Fig 6. The Polygon view of the AMMI2 model biplot for fresh root yield from 36 cassava clones grown in 17 environments.**

yield revealed the first and second principal components explained 26.5% and 17.6% and accounted for 4.0% and 2.7% of TSS respectively (S10 Table).

The AMMI-2 biplot revealed how the genotypes and environment interact to affect fresh root yield (Fig 6). The genotypes close to each other in this biplot have similar responses to environments and conversely for genotypes that are far apart. Genotypes close to show small GEI deviations while those distant from the origin show large GEI deviations.

The mean fresh root yield (t/ha) value of cassava clones averaged over testing environments indicated that clone IITA-TMS-IBA000070 had the highest fresh root yield (37.9 t/ha) and clone TMS14F1120P0003 had the lowest yield (22.5 t/ha, S11 Table). The IPC1 and IPC2 scores signify the adaptability of a genotype over environments and the relationship between genotype and environment (S11 Table). Clones with large scores in absolute value (e.g., TMS14F1297P0019 and TMEB419) have high interactions and are unstable, whereas clones with scores close to zero (IITA-TMS-IBA000070, TMS14F1036P0007) have low interactions and are stable.

The AMMI stability value (ASV) ranged from 2.60 to 40.34, averaging 15.39 across the 36 clones. The clones IITA-TMS-IBA000070 (2.60), TMS14F1306P0020 (3.28), TMS14F1223P0007 (3.88), and TMS14F1306P0015 (5.68) had the lowest ASV values, while TMEB419 (40.34), TMS14F1297P0019 (39.21), and TMS14F1300P0008 (30.41) had the highest values (S11 Table). Combining both stability and yield performance measures into

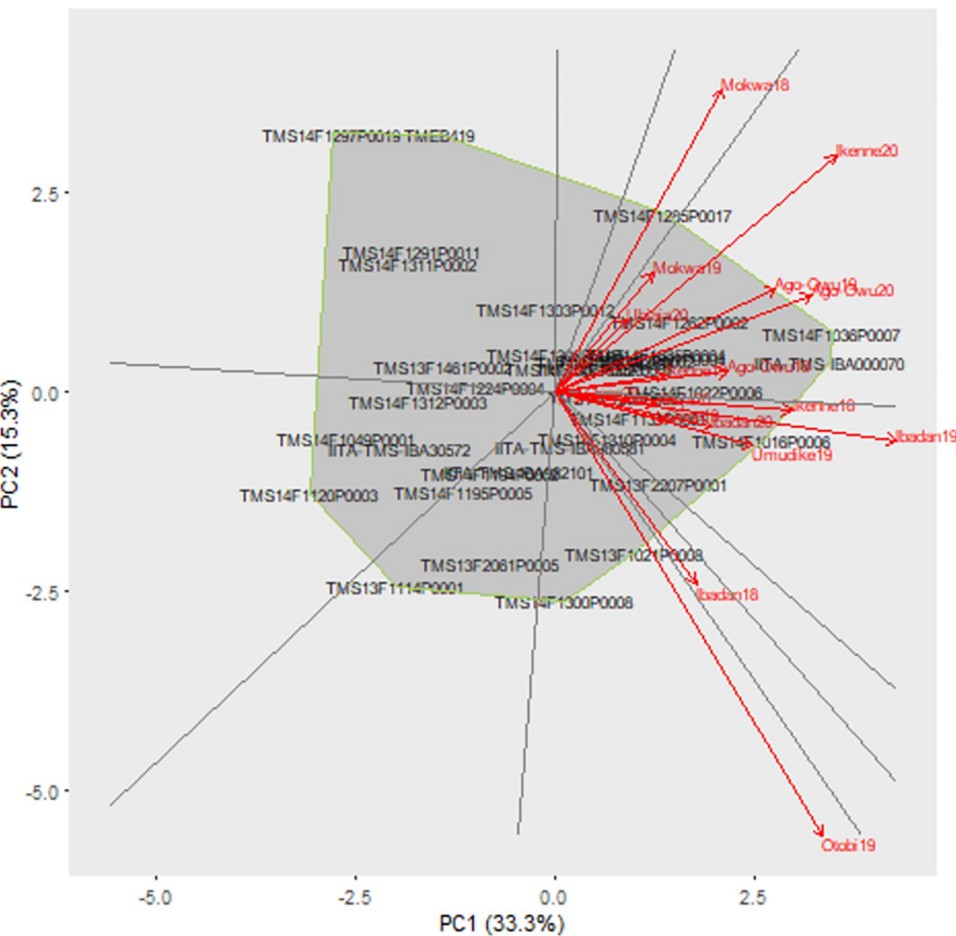

**Fig 7. The Polygon view of GGE model biplot for fresh root yield (t/ha) from 36 cassava clones grown in 17 environments.**

genotype selection index showed that IITA-TMS-IBA000070 and TMS14F1036P0007 were the two best clones (S11 Table).

## Identifying mega environments

The fitted fresh root yield values from the AMMI2 model were used to cluster 17 testing environments into six mega environments, one for each of the winning clones "IITA-TM-S-IBA000070", "IITA-TMS-IBA980581", "TMS14F1016P0006", "TMS14F1036P0007", "TMS14F1285P0017", and "TMS14F1300P0008' (S12 Table). The clones IITA-TMS-IBA000070 and TMS14F1016P0006 had broad adaptation to eight and four environments, respectively. However, clones IITA-TMS-IBA980581, TMS14F1285P0017, and TMS14F1300P0008 had specific adaptation to environments Abuja20, Mokwa18, and Ibadan18, respectively. TMS14F1036P0007 was the best clone in environments Ago-Owu19 and Ibadan20.

## GGE analysis

The GGE model showed a significant main effect of environment and combined genotype and genotype by environment interaction effect (P < = 0.001) for the observed traits (S13 Table). The partition of TSS which includes sum of squares (SS) of environment and genotype and

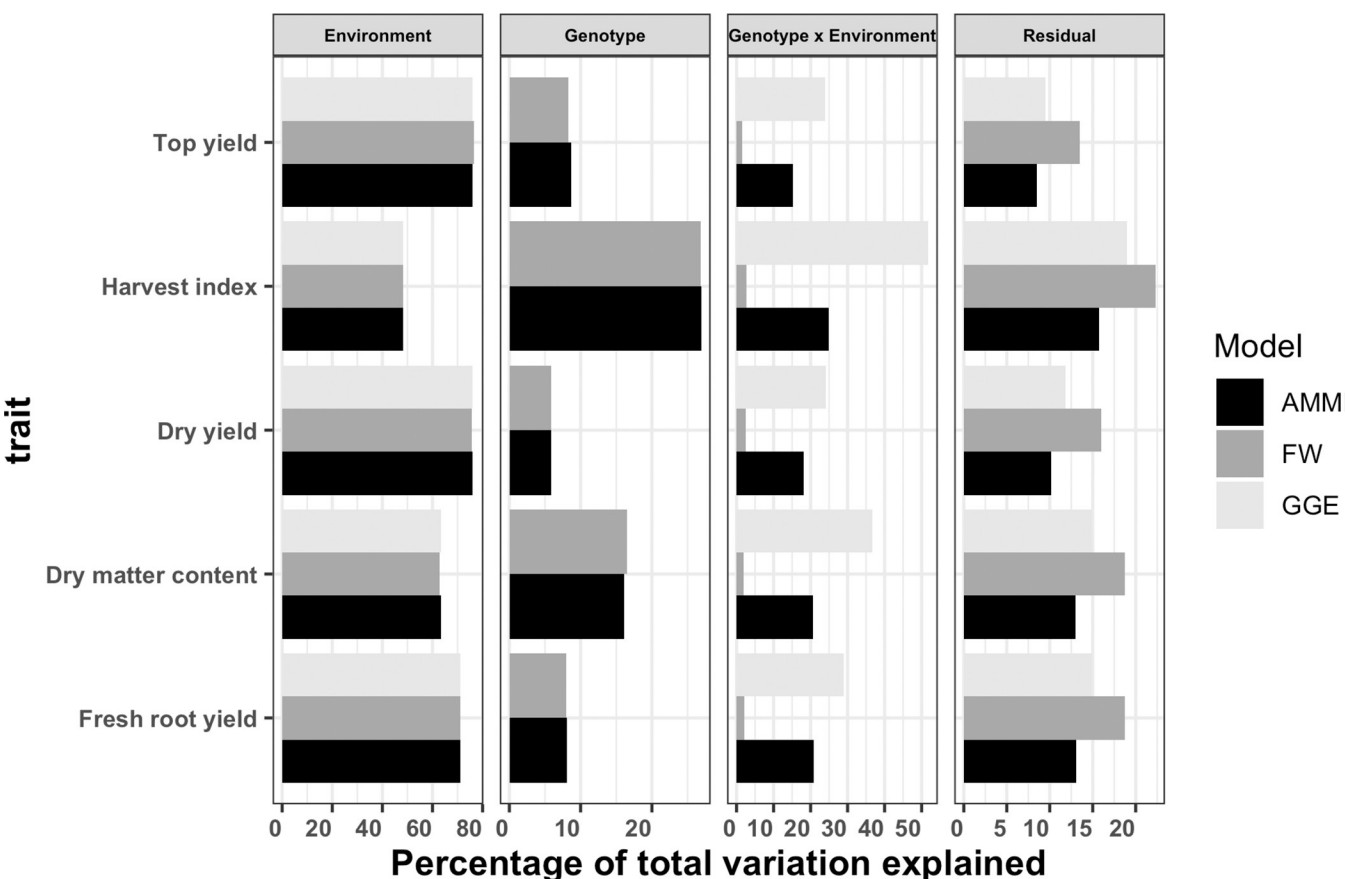

**Fig 8. Percentage of total variation captured by each factor from fitting additive main effect and multiplicative interaction (AMMI), Finlay Wilkinson (FW), and genotype and genotype by environment (GGE) models to yield related traits on 36 elite cassava clones evaluated in 17 environments.** Note that the variation attributed to genotype x environment factor for GGE model includes genotypic variance.

genotype by environment interaction indicated that environment explained a larger percentage of variation for all observed traits relative to GGE component except for harvest index. The variation explained by the GGE component ranged from 23.9% (top yield) to 51.8% (harvest index). For fresh root yield, the first and second IPCs accounted for 9.6% and 4.4% of TSS and explained 33.3% and 15.3% of GGE variation, respectively. For dry matter content, the first two IPCs captured 17.4% and 4.3% of TSS and explained 47.6% and 11.8% of GGE variation. For the top yield, the first two IPCs explained 11.3% and 3.1% of TSS and captured 47.4% and 12.8% of GGE variation.

GGE biplots based on symmetric scaling of genotype and environment were used to estimate the pattern of environments in relation to genotypes (Fig 7). Symmetric scaling is a scaling method that splits the singular value symmetrically between genotype and environment while visualizing the which-won-where pattern of the MET data. The first principal component of environment had both negative and positive scores indicating a difference in yield performance across environments resulting in cross-over GEI.

The three models revealed that the environment effect accounted for almost the same percentage of total phenotypic variation for the observed traits (Fig 8). Likewise, the genotypic effect of FW and AMMI models explained nearly the same percentage of total phenotypic variation for each measurable trait. The interaction factor of GGE model includes main effect of genotype and genotype by environment interact resulting in a larger percentage of total

**Table 2. Correlation coefficient (r) of environment specific BLUPs with all target population of environment (TPE) and environment-specific heritability ($H^2$) based on the Cullis method [34] for fresh root yield (t/ha).**

| Environment | r | $H^2$ | Rank r | Rank $H^2$ | Sum ranks |
|---|---|---|---|---|---|
| Ago-Owu19 | 0.73 | 0.80 | 1 | 2 | 3 |
| Ikenne18 | 0.69 | 0.82 | 2 | 1 | 3 |
| Ibadan19 | 0.68 | 0.73 | 4 | 5 | 9 |
| Ago-Owu18 | 0.56 | 0.74 | 9 | 4 | 13 |
| Ikenne20 | 0.67 | 0.68 | 5 | 8 | 13 |
| Onne19 | 0.67 | 0.69 | 6 | 7 | 13 |
| Ago-Owu20 | 0.68 | 0.61 | 3 | 11 | 14 |
| Mokwa18 | 0.47 | 0.74 | 12 | 3 | 15 |
| Ibadan20 | 0.61 | 0.51 | 7 | 15 | 22 |
| Umudike19 | 0.61 | 0.56 | 8 | 14 | 22 |
| Ibadan18 | 0.33 | 0.71 | 17 | 6 | 23 |
| Ikenne19 | 0.54 | 0.57 | 10 | 13 | 23 |
| Mokwa19 | 0.46 | 0.61 | 13 | 10 | 23 |
| Ubiaja20 | 0.42 | 0.62 | 15 | 9 | 24 |
| Otobi19 | 0.44 | 0.58 | 14 | 12 | 26 |
| Abuja20 | 0.49 | 0.27 | 11 | 16 | 27 |
| Onne20 | 0.40 | 0.26 | 16 | 17 | 33 |

phenotypic variation in comparison to other models. The GEI component of AMMI captured larger percentage of variation than FW, which explained relatively low variation for all the traits. For all the observed traits, the residual term of the AMMI model was the lowest while for FW it was the highest (Fig 8).

## Cultivar superiority index

Mean performance and index values of cultivar-superiority stability estimates were presented for fresh root yield, dry matter content, and dry yield to assess genotypes' stability across the testing environments (S14 Table). Among the 36 clones, 19 had a mean fresh root yield above the grand mean of 29.5 t/ha. The remaining clones had an average fresh root yield below the grand mean. Consequently, clones with above mean performance and are stable by the outcome of these stability measures are desirable.

Superiority index value $P_i$ is defined as the deviation of the ith genotype relative to the genotype with maximum performance in each environment. The top-ranked five stable clones for fresh root yield with lowest $P_i$ value included IITA-TMS-IBA000070, TMS14F1036P0007, TMS14F1016P0006, TMS14F1262P0002, and TMS14F1035P0004 [33]. These clones also have relatively high fresh root yield above grand average yield of 29.5 t/ha and their corresponding dry matter ranged from 31.3% for TMS14F1016P0006 to 37.4% for TMS14F1035P0004 (S4 Fig and S14 Table). The top-ranked five clones for dry matter content include TMS14F1035P0004, TMS14F1306P0015, TMS14F1291P0011, TMS14F1195P0005, and TMS14F1049P0001 (S5 Fig and S14 Table). The superiority index associated to dry yield showed the best clones for both fresh root yield and dry matter content. The top-ranked five clones were TMS14F1036P0007, IITA-TMS-IBA000070, TMS14F1035P0004, TMS14F1262P0002, and TMS13F2207P0001 (S6 Fig and S14 Table).

## Representative of target population of environments

The correlation coefficient of each environment's BLUPs with genotypic BLUPs of all environments in the TPE for fresh root yield ranged from 0.33 (Ibadan18) to 0.73 (Ago-Owu19) with

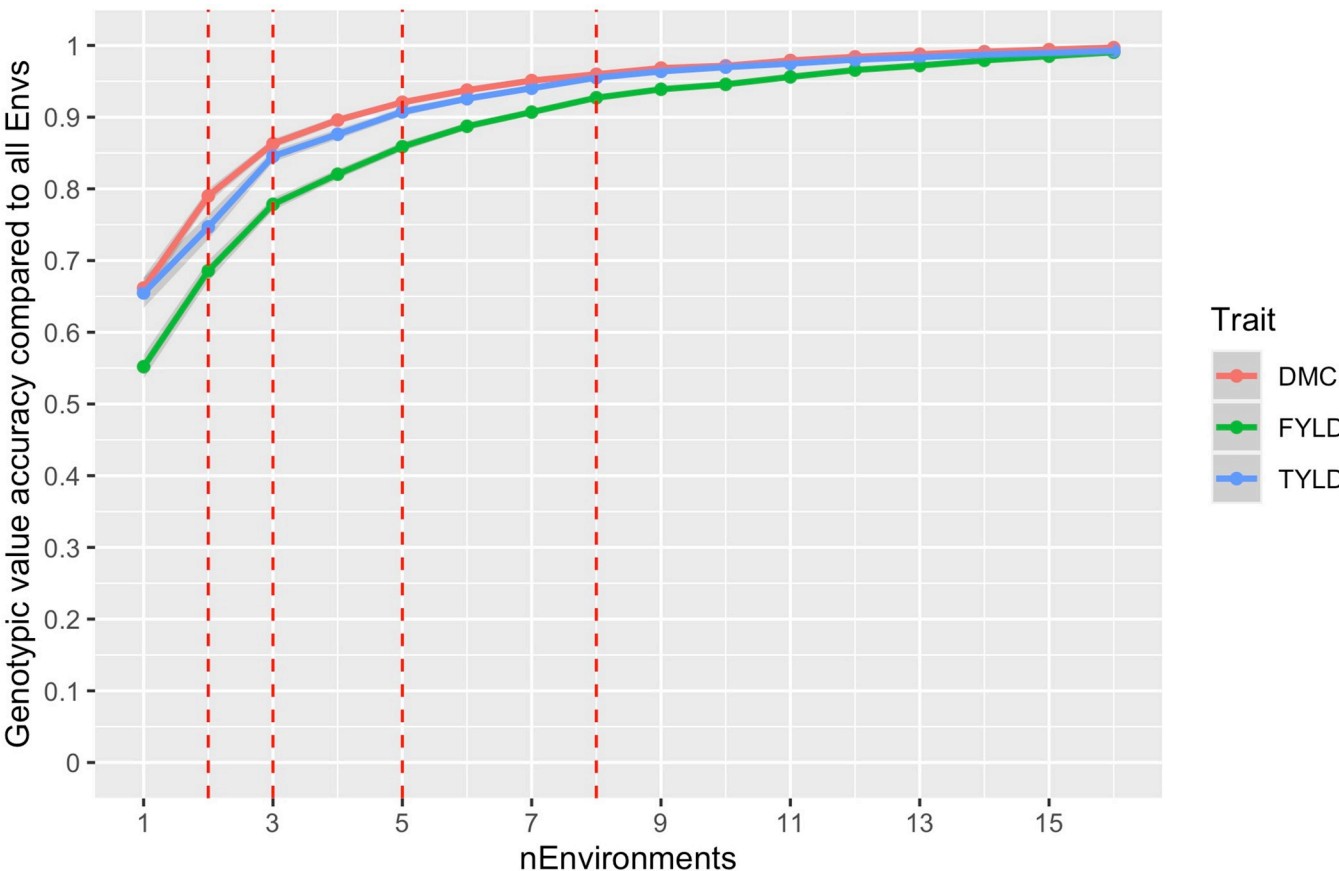

**Fig 9. Estimated genotypic value accuracy against the number of sampling environments for dry matter content (DMC), fresh root yield (FYLD), and top yield (TYLD).**

corresponding heritability estimates of 0.71 and 0.80, respectively (Table 2). The top ranked 5 environments showing high correlation with TPE, and high heritability estimate include Ago-Owu19 (0.73, 0.80), Ikenne18 (0.69, 0.82), Ibadan19 (0.68, 0.73), Ago-Owu18 (0.56, 0.74), and Ikenne20 (0.67,0.68) (Table 2).

As for dry matter content, the environments revealed a higher range of correlation coefficient with TPE relative to fresh root yield varying from Onne20 (0.48) to Ikenne20 (0.85) with corresponding heritability estimate of 0.57 and 0.88, respectively (S15 Table). The top ranked 5 environments to represent the TPE for showing high correlation and high heritability included Ikenne20 (0.85, 0.88), Ikenne18 (0.79, 0.78), Onne19 (0.72, 0.87), Ubiaja20 (0.73, 0.82), and Umudike (0.78, 0.77). For top yield, there was higher variability in the correlation coefficient with TPE ranging from 0.14 (Mokwa19) to 0.83 (Ubiaja20) with heritability estimates of 0.32 and 0.71, respectively (S16 Table). The top ranked 5 environments to represent the TPE were Ikenne19 (0.82, 0.81), Otobi19 (0.83, 0.78), Ikenne18 (0.79, 0.82), Ago-Owu20 (0.79, 0.76), and Ubiaja20 (0.83, 0.71).

A line graph provides further insights into the number of environment(s) that is likely to be sampled to represent TPE and their corresponding genotypic value accuracy compared to all the environment for fresh root yield, dry matter content, and top yield (Fig 9). Fresh root yield's genotypic value accuracy is the lowest as revealed in Fig 9, sampling of five (5) environments is likely to represent TPE where fresh root yield has an approximate genotypic value accuracy of 0.84 lower than dry matter content and top yield with genotypic value accuracy of

0.92 and 0.91 respectively. Regardless the number of environments sampled, the breeding value accuracy of fresh root yield is lower than that of dry matter content and top yield.

The relatedness among the testing TPE for fresh root yield revealed the grouping of the testing TPE into three cluster groups such that environments within a cluster are more similar and dissimilar from environments in another cluster (S7 Fig). As for dry matter content, the TPEs were grouped into 4 clusters (S8 Fig). However, 6 cluster groups of TPE were identified for top yield out of which 3 clusters have one environment each (S9 Fig).

## Discussion

This study demonstrated the application of classical ANOVA in a linear mixed model framework and linear-bilinear models such as Finlay-Wilkinson, additive main effect and multiplicative interaction model, and genotype plus genotype-environment models towards identifying stable clones, mega-environments and environments representative of the TPE.

The large sum of squares and significant effect of environment on the observed agronomic traits as shown by FW, AMMI, and GGE models demonstrated that the field trials were conducted under diverse environmental conditions causing variation in cassava clones yield and other yield-related traits. The significant variation of the GEI effect found for the observed agronomic traits indicated that genotype and environment main effects cannot independently capture all the variation observed. This resulted in the diverse performance of the clones in the testing environments. This variation therefore requires to examine GEI and assess the stability of clones.

We found that the AMMI model attributed the most significant percentage of treatment sum of squares to the environment for the observed traits and that the main effects of genotype, environment and their interaction were significant for all observed traits. This finding was similar to Dixon et al. [36] who reported significant effects of genotype, environment and GEI. However, this finding was contrary to Tumuhimbise et al. [2] who reported that genotype accounted for the largest percentage of the treatment sum of squares (48.5%). The disparity in the result may be because our study evaluated 36 clones in 17 environments compared to 12 clones in three environments in Tumuhimbise et al. [2]. Also contrary to our findings, Jiwuba et al. [37] reported that GEI accounted for the largest percentage of the treatment sum of squares (43.80%) in their study where 60 genotypes were evaluated over six environments.

As for top yield, the environment captured the largest percentage of total variability (76.1%) from the AMMI model. This was contrary to findings from Jiwuba et al. [37] who reported that the environment accounted for the lowest percentage of total variation (11.9%) for the biomass. The disparity in the result may be because they evaluated six environments in their study compared to 17 environments in our study. Unlike fresh root yield and top yield, for dry matter content, the percentage of total sum of squares attributed to G (16.1%) was relatively close to GEI (20.6%). However, all the linear bilinear models explored in this study revealed that the environment accounted for much greater DMC variation than genotypic effect. This may be because this is a UYT study, so that clones have already been strongly selected, so that genetic variability is reduced. In contrast, Benesi et al. [38] reported that genotypic influence on dry matter content is much higher than that of the environment.

Like the FW model, AMMI revealed a significant genotypic effect for the observed agronomic traits, signifying the presence of genetic variation in IITA cassava germplasm. This is similar to what Nduwumuremyi et al. [39] who reported about the existence of significant genetic variation in Rwandan germplasm.

The limitation of classical ANOVA is that it does not provide insight into the complex pattern of GEI, which necessitates further use of linear bilinear models. The strength of AMMI

and GGE models is that they concurrently visualize genotypes and environments using biplots that expedite the interpretation of GEI. In biplots, a genotype in the vicinity of an environment with a large IPC score is expected to display a higher performance in that environment than its mean performance, and conversely for genotypes located far from that environment on the biplot.

## Conclusion

The classical statistical methods used in this study found highly significant genotype-by-environment interaction, a major challenge confronting cassava breeders in the course of breeding for high yielding and stable varieties. We were able to identify high yielding clones with broad or specific adaptation across the testing environments. There were six mega-environments identified from 17 testing environments as a function of winning genotypes. The outcomes from this study provide further insight for other breeders that intend to embark on similar tasks. The research objectives outlined in this study have been achieved based on the results obtained.

The Finlay-Wilkinson, AMMI, and GGE are fixed effect models, and they may not be an appropriate approach to use when estimating quantitative genetic parameters in the presence of unbalanced data and/or when jointly analyzing heterogeneous trial designs. Such circumstances require a mixed model approach where different variance covariance structures can be explored. In addition, these models assumed homogeneity of error variances across the testing environments which may be incorrect as error variances were heterogeneous as revealed through likelihood ratio tests. None of these linear bilinear models can account for relatedness among the genotypes, e.g., using relatedness matrices from pedigree and/or molecular data.

Though the same clones were evaluated across the testing environments (trials or location by year combinations), there were locations (Abuja, Otobi, Umudike, and Ubiaja) where this study was carried out in just one out of three cropping seasons resulting in an unbalanced data structure. Therefore, the outcome of delineating the testing environments into mega-environments may be misleading. To ensure having well-defined mega-environments, it would be advisable to have more than one cropping season of data from all locations. To get a clearer picture of locations that are representative of the TPE, studies should also require several years of historical data. Finally, to get better understand the factors influencing the GEI, soil and weather data are needed explicitly.

## Supporting information

**S1 Fig. This is the scatter plot of coefficient of variation and heritability.**
(PDF)

**S2 Fig. This is the scatter plot of experimental accuracy and heritability.**
(PDF)

**S3 Fig. This is the boxplot showing distribution of observed traits for some parameter estimates.**
(PDF)

**S4 Fig. This is the scatter plot of cultivar superiority index and mean fresh root yield.**
(PDF)

**S5 Fig. This is the scatter plot of cultivar superiority index and mean dry matter content.**
(PDF)

**S6 Fig. This is the scatter plot of cultivar superiority index and mean dry yield.**
(PDF)

**S7 Fig. This is clustering of environments based on genotypic BLUPS of fresh root yield.**
(PDF)

**S8 Fig. This is clustering of environments based on genotypic BLUPS of dry matter content.**
(PDF)

**S9 Fig. This is clustering of environments based on genotypic BLUPS of top yield.**
(PDF)

**S1 Table. This is the summary statistics of individual trials.**
(XLSX)

**S2 Table. This is the likelihood ratio test of absence versus presence of GEI.**
(XLSX)

**S3 Table. This is the likelihood ratio test of homogeneity versus heterogeneity of error variances.**
(XLSX)

**S4 Table. This is the ANOVA table showing the variance component estimates.**
(XLSX)

**S5 Table. This is ANOVA table showing the partition of GEI variance component.**
(XLSX)

**S6 Table. This is the ANOVA table resulting from Finlay-Wilkinson (FW) model.**
(XLSX)

**S7 Table. This is the genotypes ranking based on sensitivities values from FW model.**
(XLSX)

**S8 Table. This is minimum, median, maximum, and variance of MSE from FW model.**
(XLSX)

**S9 Table. This is minimum, median, maximum, and variance of slopes from FW model.**
(XLSX)

**S10 Table. This is the combined ANOVA table resulting from AMMI model.**
(XLSX)

**S11 Table. This is the ranking of genotypes based on AMMI stability value and genotype selection index.**
(XLSX)

**S12 Table. This is the table showing mega-environment based on AMMI2 model.**
(XLSX)

**S13 Table. This is the combined ANOVA table resulting from GGE model.**
(XLSX)

**S14 Table. This is the assessment of stability of genotypes based on cultivar superiority index.**
(XLSX)

**S15 Table. This is the ranking of testing environments for dry matter content based on correlation and Cullis heritability.**
(XLSX)

**S16 Table. This is the ranking of testing environments for top yield based on correlation and Cullis heritability.**
(XLSX)

## Acknowledgments

The authors extend their appreciation to the dedicated staff of the International Institute of Tropical Agriculture (IITA) of the cassava breeding program that assisted with the fieldwork, screen house evaluation and laboratory analyses. We also thank the referees for comments and suggestions that have improved the manuscript.

## Author Contributions

**Conceptualization:** Moshood A. Bakare, Siraj Ismail Kayondo, Peter Kulakow, Ismail Yusuf Rabbi, Jean-Luc Jannink.

**Data curation:** Moshood A. Bakare, Siraj Ismail Kayondo, Ismail Yusuf Rabbi.

**Formal analysis:** Moshood A. Bakare, Siraj Ismail Kayondo, Marnin D. Wolfe, Ismail Yusuf Rabbi.

**Funding acquisition:** Peter Kulakow, Chiedozie Egesi, Ismail Yusuf Rabbi, Jean-Luc Jannink.

**Investigation:** Moshood A. Bakare, Siraj Ismail Kayondo, Cynthia I. Aghogho.

**Methodology:** Moshood A. Bakare, Siraj Ismail Kayondo, Cynthia I. Aghogho, Ismail Yusuf Rabbi, Jean-Luc Jannink.

**Project administration:** Elizabeth Y. Parkes, Peter Kulakow, Chiedozie Egesi, Ismail Yusuf Rabbi, Jean-Luc Jannink.

**Supervision:** Peter Kulakow, Ismail Yusuf Rabbi, Jean-Luc Jannink.

**Visualization:** Moshood A. Bakare.

**Writing – original draft:** Moshood A. Bakare, Cynthia I. Aghogho, Marnin D. Wolfe, Jean-Luc Jannink.

**Writing – review & editing:** Moshood A. Bakare, Siraj Ismail Kayondo, Cynthia I. Aghogho, Marnin D. Wolfe, Peter Kulakow, Ismail Yusuf Rabbi, Jean-Luc Jannink.

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
