## [Decision Letter · Decision Letter 0]

7 Mar 2022

PONE-D-21-40030Exploring genotype by environment interaction on cassava yield and yield related traits using classical statistical methodsPLOS ONE

Dear Dr. Jannink,

Thank you for submitting your manuscript to PLOS ONE. After careful consideration, we feel that it has merit but does not fully meet PLOS ONE’s publication criteria as it currently stands. Therefore, we invite you to submit a revised version of the manuscript that addresses the points raised during the review process.

We look forward to receiving your revised manuscript.

Kind regards,

Muhammad Amjad Ali, PhD

Academic Editor

PLOS ONE

Journal Requirements:

Additional Editor Comments:

Dear Authors,

We have received the review reports on your manuscript "Exploring genotype by environment interaction on cassava yield and yield related traits using classical statistical methods

" from the renowned reviewers in the field.

As you can see the comments, they have asked for revisions.

Please go through the comments and revise the manuscript inline with the comments and suggestions.

Regards,

Muhammad Amjad Ali

Reviewers' comments:

Reviewer's Responses to Questions

**Comments to the Author**

1. Is the manuscript technically sound, and do the data support the conclusions?

Reviewer #1: Yes

Reviewer #2: Partly

2. Has the statistical analysis been performed appropriately and rigorously? 

Reviewer #1: Yes

Reviewer #2: I Don't Know

3. Have the authors made all data underlying the findings in their manuscript fully available?

Reviewer #1: Yes

Reviewer #2: Yes

4. Is the manuscript presented in an intelligible fashion and written in standard English?

Reviewer #1: Yes

Reviewer #2: Yes

5. Review Comments to the Author

Reviewer #1: I have reviewed the manuscript # PONE-D-21-40030,

Title “Exploring genotype by environment interaction on cassava yield and yield related traits using classical statistical methods”.

I am glad to express that the manuscript is well written, and the message is clearly presented. The standard of language is also good. The introduction is good and clearly describes the background of the study. The analytical methods used, and the conclusions drawn are satisfactory.

Although the research seems more of regional importance rather than international, but I think it’s still good enough to contribute to the global knowledge. The statistical methods also look conventional. But keeping in mind the current situation of lack of facilities for advanced genomics tools for every researcher, the guidelines provided for conventional methods will be of valuable importance to make genetic gains on a very important problem of GxE in cassava.

Having said that, I still want to suggest some improvements to improve the manuscript, such as

1. There are a few minor typo errors, I have mentioned them as sticky notes on the attached pdf document.

2. In materials and methods section, please add some salient features/background information of the clones/genotypes used in the study, to indicate why they were selected for the experiment. The relevant information can be added to Table 1, the table also needs improvement in formatting.

3. In materials and methods section, it is suggested that instead of showing coordinates of study sites in tabular form (Table 2) they can be displayed in a form of a map with study sites displayed as dots.

4. Similarly in materials and methods section, the weather data in Table 2 can be displayed in a graphical form for a clear understanding of readers.

5. The experimental material is designated with 2 different terms in various section of the manuscript such as

a. “clones” in abstract and material & methods sections

b. “genotypes” in results & discussion sections. It should be consistent throughout

6. I suggest that conclusion should be reconsidered so that it can show that based on the results all the objectives of the study were achieved.

To summarise it, I recommend that the manuscript may be considered suitable for publication with minor revisions.

Reviewer #2: In this study, author have investigated the dissection of the existing patterns of GEI using linear-bilinear models such as Finlay-Wilkinson, additive main effect and multiplicative interaction, and genotype and genotype by environment interaction models are critical in defining the target population of environments for future testing, selection, and advancement. Study is well designed but i have some suggestion

Methods and results section is very long. I suggest you should make it little short.

While, discussion section is very short. Please discuss results in detail and add appropriate references.

6. PLOS authors have the option to publish the peer review history of their article (what does this mean?). If published, this will include your full peer review and any attached files.

Reviewer #1: **Yes: **Shahid Iqbal Awan

Reviewer #2: No

---

## [Author Response · Author response to Decision Letter 0]

21 Apr 2022

1. Is the manuscript technically sound, and do the data support the conclusions?

Reviewer #1: Yes

Reviewer #2: Partly

2. Has the statistical analysis been performed appropriately and rigorously?

Reviewer #1: Yes

Reviewer #2: I Don't Know

3. Have the authors made all data underlying the findings in their manuscript fully available?

Reviewer #1: Yes

Reviewer #2: Yes

4. Is the manuscript presented in an intelligible fashion and written in standard English?

Reviewer #1: Yes

Reviewer #2: Yes

5. Review Comments to the Author

Reviewer #1: I have reviewed the manuscript # PONE-D-21-40030,

Title “Exploring genotype by environment interaction on cassava yield and yield related traits using classical statistical methods”.

I am glad to express that the manuscript is well written, and the message is clearly presented. The standard of language is also good. The introduction is good and clearly describes the background of the study. The analytical methods used, and the conclusions drawn are satisfactory.

Although the research seems more of regional importance rather than international, but I think it’s still good enough to contribute to the global knowledge. The statistical methods also look conventional. But keeping in mind the current situation of lack of facilities for advanced genomics tools for every researcher, the guidelines provided for conventional methods will be of valuable importance to make genetic gains on a very important problem of GxE in cassava.

Having said that, I still want to suggest some improvements to improve the manuscript, such as

1. There are a few minor typo errors, I have mentioned them as sticky notes on the attached pdf document.

Response

The typo errors have been rectified. Meanwhile, the first sticker note is not the author name italicized but the botanical name of cassava (Manihot esculenta Crantz) as in line 72. Line 172 abbreviation IITA has been rectified to Internatiional Institute of Tropical Agriculture

2. In materials and methods section, please add some salient features/background information of the clones/genotypes used in the study, to indicate why they were selected for the experiment. The relevant information can be added to Table 1, the table also needs improvement in formatting.

Response

We have provided short description about where the clones were derived from and how they were screened for diseases, quality and agronomic traits of interest over different breeding stages of evaluation. This can be found from Line 174-178 of revised document. In addition, we added the cycle each clone belongs and the year of cloning as in Table 1 line 182. We further formatted the tables by bolding the table title and aligning the text within each column.

3. In materials and methods section, it is suggested that instead of showing coordinates of study sites in tabular form (Table 2) they can be displayed in a form of a map with study sites displayed as dots.

Response

We have reproduced a Nigerian map as Fig 1 showing the testing locations for the field trials and the agro-ecological zone each belongs to as in line 202 of revised manuscript with track changes.

4. Similarly in materials and methods section, the weather data in Table 2 can be displayed in a graphical form for a clear understanding of readers.

Response

A line graph named Fig 2 has been produced showing the trend of average minimum and maximum temperature, total precipitation, and mean relative humidity across the testing environments as in line 221 of revised manuscript with track changes.

5. The experimental material is designated with 2 different terms in various section of the manuscript such as

a. “clones” in abstract and material & methods sections

b. “genotypes” in results & discussion sections. It should be consistent throughout

Response

Every occurrence of “cassava genotypes” which represents the treatment in the abstract, material and methods, and result and discussion sections have been replaced with the word “clones” to be consistent.

6. I suggest that conclusion should be reconsidered so that it can show that based on the results all the objectives of the study were achieved.

Response

We have updated the conclusion to reflect that all research objectives as outlined in this study were achieved based on the outcome of the study. This can be found as from line 1039 to 1045 of revised manuscript with track changes.

.

To summarise it, I recommend that the manuscript may be considered suitable for publication with minor revisions.

Reviewer #2: In this study, author have investigated the dissection of the existing patterns of GEI using linear-bilinear models such as Finlay-Wilkinson, additive main effect and multiplicative interaction, and genotype and genotype by environment interaction models are critical in defining the target population of environments for future testing, selection, and advancement. Study is well designed but i have some suggestion

Methods and results section is very long. I suggest you should make it little short.

While, discussion section is very short. Please discuss results in detail and add appropriate references.

Response

We revised both the materials and method and result sections by taking out some redundant statements as shown in a marked-up copy of the revised document. We have discussed the findings in this study and related them to findings from references such as Dixon et al, Tumuhimbise et al, Benesi et al, and Nduwumuremyi et al.

6. PLOS authors have the option to publish the peer review history of their article (what does this mean?). If published, this will include your full peer review and any attached files.

Do you want your identity to be public for this peer review? For information about this choice, including consent withdrawal, please see our Privacy Policy.

Reviewer #1: Yes: Shahid Iqbal Awan

Reviewer #2: No

---

## [Editor Report · Decision Letter 1]

25 Apr 2022

Exploring genotype by environment interaction on cassava yield and yield related traits using classical statistical methods

PONE-D-21-40030R1

Dear Dr. Jannink,

We’re pleased to inform you that your manuscript has been judged scientifically suitable for publication and will be formally accepted for publication once it meets all outstanding technical requirements.

Kind regards,

Muhammad Amjad Ali, PhD

Academic Editor

PLOS ONE
---

## [Editor Report · Acceptance letter]

28 Apr 2022

PONE-D-21-40030R1 

Exploring genotype by environment interaction on cassava yield and yield related traits using classical statistical methods 

Dear Dr. Jannink:

I'm pleased to inform you that your manuscript has been deemed suitable for publication in PLOS ONE. Congratulations! Your manuscript is now with our production department. 

Kind regards, 

on behalf of

Dr. Muhammad Amjad Ali 

Academic Editor

PLOS ONE